# Betacoronaviruses Differentially Activate the Integrated Stress Response to Optimize Viral Replication in Lung-Derived Cell Lines

**DOI:** 10.3390/v17010120

**Published:** 2025-01-16

**Authors:** David M. Renner, Nicholas A. Parenti, Nicole Bracci, Susan R. Weiss

**Affiliations:** 1Department of Microbiology, Perelman School of Medicine, University of Pennsylvania, Philadelphia, PA 19104, USA; dmrenner1994@gmail.com (D.M.R.); nicholas.parenti@pennmedicine.upenn.edu (N.A.P.); nicole.bracci@pennmedicine.upenn.edu (N.B.); 2Penn Center for Research on Coronaviruses and Other Emerging Pathogens, Perelman School of Medicine, University of Pennsylvania, Philadelphia, PA 19104, USA

**Keywords:** coronavirus, integrated stress response, PERK pathway, PKR pathway, SARS-CoV-2, HCoV-OC43, MERS-CoV

## Abstract

The betacoronavirus genus contains five of the seven human coronaviruses, making it a particularly critical area of research to prepare for future viral emergence. We utilized three human betacoronaviruses, one from each subgenus—HCoV-OC43 (embecovirus), SARS-CoV-2 (sarbecovirus), and MERS-CoV (merbecovirus)—, to study betacoronavirus interactions with the PKR-like ER kinase (PERK) pathway of the integrated stress response (ISR)/unfolded protein response (UPR). The PERK pathway becomes activated by an abundance of unfolded proteins within the endoplasmic reticulum (ER), leading to phosphorylation of eIF2α and translational attenuation. We demonstrate that MERS-CoV, HCoV-OC43, and SARS-CoV-2 all activate PERK and induce responses downstream of p-eIF2α, while only SARS-CoV-2 induces detectable p-eIF2α during infection. Using a small molecule inhibitor of eIF2α dephosphorylation, we provide evidence that MERS-CoV and HCoV-OC43 maximize viral replication through p-eIF2α dephosphorylation. Interestingly, genetic ablation of growth arrest and DNA damage-inducible protein (GADD34) expression, an inducible protein phosphatase 1 (PP1)-interacting partner targeting eIF2α for dephosphorylation, did not significantly alter HCoV-OC43 or SARS-CoV-2 replication, while siRNA knockdown of the constitutive PP1 partner, constitutive repressor of eIF2α phosphorylation (CReP), dramatically reduced HCoV-OC43 replication. Combining GADD34 knockout with CReP knockdown had the maximum impact on HCoV-OC43 replication, while SARS-CoV-2 replication was unaffected. Overall, we conclude that eIF2α dephosphorylation is critical for efficient protein production and replication during MERS-CoV and HCoV-OC43 infection. SARS-CoV-2, however, appears to be insensitive to p-eIF2α and, during infection, may even downregulate dephosphorylation to limit host translation.

## 1. Introduction

Protein production is critical for cellular survival and viral replication. Translational control offers the cell the chance to respond to various forms of stress that may influence proteostasis or protein quality control. Mammals have evolved an elegant system, termed the integrated stress response (ISR), for detecting and responding to these perturbations and limiting translation while attempting to restore homeostasis [1].

The ISR is composed of four kinases that all converge on the phosphorylation of serine 51 of the alpha subunit of eukaryotic initiation factor 2 (eIF2α). These proteins share highly conserved kinase domains but detect and respond to different types of cellular stress. General control nonderepressible 2 (GCN2), the most ancient ISR kinase conserved down to budding yeast, responds to amino acid starvation, ribosome stalling [1], and ribosome collisions [2]. Heme-regulated inhibitor (HRI) senses and responds to heme starvation and oxidative stress [1], and has recently been tied to mitochondrial stress [3]. Protein kinase R (PKR) binds to double-stranded RNA (dsRNA), a replication intermediate of RNA and some DNA viruses, making the ISR partly overlap with innate immunity and the interferon response [1,4]. The fourth kinase, PKR-like ER kinase (PERK), is a transmembrane protein residing in the ER. The luminal domain of PERK is bound by binding immunoglobulin proteins (BiP), a chaperone within the ER lumen. As a consequence of ER stress, BiP dissociates from PERK, inducing PERK activation and phosphorylation of eIF2α, which limits translation and the influx of nascent peptides into the ER. PERK, along with inositol requiring enzyme 1α (IRE1α) and activating transcription factor 6 (ATF6), also constitutes part of the unfolded protein response (UPR), which serves to sense and respond to stress within the ER [5]. Thus, the ISR serves a central role in detecting and responding to stress within mammalian cells and overlaps extensively with other more specific stress pathways.

Phosphorylation of eIF2α limits the availability of the eIF2:GTP:Met-tRNA_i_^Met^ ternary complex, thus limiting cap-dependent translation [1,6]. While the translation of most mRNAs is limited when eIF2α is phosphorylated, a subset of mRNAs are translated more efficiently under these conditions. Certain response factors, such as activating transcription factor 4 (ATF4), have upstream open reading frames (uORFs) in the 5′ end of their mRNAs. During homeostatic conditions, ribosomes preferentially initiate on these uORFs, synthesizing short, abortive peptides rather than the true coding sequence. When ternary complex abundance is low, translation initiation is slowed, allowing ribosomes to scan through uORFs or reinitiate on the correct ORF [7]. ATF4 is translated under conditions of translation attenuation and serves as the master transcriptional regulator of the ISR. ATF4 induces a transcriptional cascade aimed at alleviating stress and restoring proteostasis. If the stress is too great or cannot be resolved, the ISR can also induce pro-apoptotic genes such as the C/EBP homologous protein (CHOP) to destroy chronically stressed cells [8,9].

If the stress has been resolved, eIF2α must be dephosphorylated to restore full translational capacity. Dephosphorylation is catalyzed by protein phosphatase 1 (PP1), which is directed to p-eIF2α by two different regulatory subunits [10]. Constitutive repressor of eIF2α phosphorylation (CReP) directs continuous, low-level dephosphorylation of eIF2α under all conditions [11]. This protein serves the role of maintaining a minimal concentration of the ternary complex within the cell at all times so that low levels of translation are maintained to respond to stress [1]. Growth arrest and DNA damage-inducible 34 (GADD34) is an inducible uORF-regulated PP1 interacting partner that is induced downstream of ATF4 and highly expressed with prolonged eIF2α phosphorylation [12]. This serves as a negative feedback loop within the ISR, promoting robust eIF2α dephosphorylation to restore translation and inhibit GADD34’s own induction if proteostasis has been restored [13] (Figure 1).

One function of the ISR is to detect and combat viral infection, which has the potential to activate multiple ISR kinases depending on the viral replication cycle. Coronaviruses (CoVs) are large, single-stranded, and positive-sense RNA viruses that establish infection within the host’s ER. To date, there are seven known human CoVs spanning two genera: alpha- and betacoronavirus. In the 21st century, three highly lethal human CoVs have emerged: severe acute respiratory syndrome (SARS)-CoV in 2002, Middle East respiratory syndrome (MERS)-CoV in 2012, and SARS-CoV-2 in 2019. All of these viruses belong to the betacoronavirus genus but to different subgenera. SARS-CoV and SARS-CoV-2 are sarbecoviruses, while MERS-CoV is a merbecovirus. Furthermore, two common cold-causing human coronaviruses—HCoV-OC43 and HCoV-HKU1—fall into a third subgenus, embecoviruses [14]. During infection, CoVs vastly remodel the host’s ER, form viral replication factories in ER-derived double-membrane vesicles (DMVs) [15,16,17], and produce dsRNA as a replication intermediate [18]. Additionally, three viral structural glycoproteins (spike, membrane, and envelope) are membrane-embedded and require trafficking through the ER, causing the ER to be flooded with viral proteins. Lastly, new viral particles form by budding into the ER–Golgi intermediate complex (ERGIC), thus depleting cellular membranes as new enveloped virions bud from the cell by exocytosis [14]. Thus, we hypothesized that coronavirus infection triggers the necessary stress stimuli to induce PKR and PERK activation during infection.

Viral interactions with the ISR have been extensively reported, particularly interactions with PKR. We have previously demonstrated that during infection, MERS-CoV and SARS-CoV-2 interact differently with PKR. MERS-CoV encodes efficient antagonists of PKR activation [19,20] while SARS-CoV-2 induces p-PKR and p-eIF2α during infection [18]. Indeed, many viruses encode antagonists of PKR to limit translational shutdown during infection [21,22,23,24,25], while others have been reported to activate multiple kinases within the ISR [18,26,27]. Some viruses, such as the alphacoronavirus transmissible gastroenteritis virus (TGEV) [28], herpes simplex 1 (HSV-1) [29], and African swine fever virus (ASFV) [30] even encode GADD34-analogous viral proteins that maintain translation within the infected cell. However, coronavirus interactions with other ISR kinases, such as PERK, have remained relatively unexplored.

Here, we compared three human betacoronaviruses from different subgenera—HCoV-OC43, SARS-CoV-2, and MERS-CoV [31]—and explore their interactions with the ISR. We focused specifically on the activation of the ISR kinases PERK and PKR, the downstream effects on p-eIF2a, and the role of GADD34 and CReP during infection. We found that all three viruses activate PERK during infection, but only SARS-CoV-2 induces p-eIF2α. Despite this, all of these viruses induce downstream signaling events of the ISR, including GADD34 upregulation. Utilizing chemical inhibitors of GADD34 and CReP [32], along with genetic ablation, we show that HCoV-OC43 relies primarily on CReP to maintain eIF2α dephosphorylation and efficient viral replication [1]. Disruption of eIF2α dephosphorylation is detrimental to MERS-CoV and, to a greater extent, HCoV-OC43 protein production and replication, but not SARS-CoV-2. Interestingly, our data suggest that SARS-CoV-2 may hinder eIF2α dephosphorylation by limiting CReP and GADD34 expression. Our findings elucidate the role of the ISR and p-eIF2α in controlling different human coronavirus infections and establish PP1-mediated eIF2α dephosphorylation [33] as a host-directed therapeutic target for some human betacoronaviruses.

## 2. Materials and Methods

**Cell Lines**. Human A549 cells (ATCC CCL-185) and its derivatives were cultured in RPMI 1640 (Gibco, Paisley, Scotland, United Kingdom, catalog no. 11875) supplemented with 10% fetal bovine serum (FBS), 100 U/mL penicillin, and 100 mg/mL streptomycin (Gibco catalog no. 15140). African green monkey kidney Vero cells (E6) (ATCC CRL-1586) and VeroCCL81 cells (ATCC CCL-81) were cultured in Dulbecco’s modified Eagle’s medium (DMEM; Gibco catalog no. 11965) supplemented with 10% FBS, 100 U/mL of penicillin, 100 mg/mL streptomycin, 50 mg/mL gentamicin (Gibco catalog no. 15750), 1 mM sodium pyruvate (Gibco catalog no. 11360), and 10 mM HEPES (Gibco catalog no. 15630). Human Calu-3 cells (ATCC HTB-55) were cultured in DMEM supplemented with 20% FBS without antibiotics. A549^DPP4^ [19] and A549^ACE2^ [18] cells were generated as described previously. CRISPR-Cas9 knockout cell lines were generated using lentiviruses. Lentivirus stocks were generated by using lentiCRISPR v2 (Addgene #42230) with single guide RNA (sgRNA) targeting GADD34 (AAGGTTCTGATAAGAACCCA) or scrambled sequence (TTCTCCGAACGTGTCACGT).

**Viruses**. SARS-CoV-2 (USA-WA1/2020) was obtained from BEI Resources, NIAID, NIH, and propagated in VeroE6-TMPRSS2 cells. The genomic RNA was sequenced and found to be identical to that of GenBank version no. MN985325.1. Recombinant MERS-CoV was described previously [20] and propagated in VeroCCL81 cells. SARS-CoV-2 and MERS-CoV infections were performed in a biosafety level 3 (BSL-3) laboratory under BSL-3 conditions, using appropriate and approved personal protective equipment and protocols. HCoV-OC43 was obtained from ATCC (VR-1558) and grown and titrated on VeroE6 cells at 33 °C.

**Viral growth kinetics and titration**. SARS-CoV-2, MERS-CoV, and HCoV-OC43 infections and plaque assays were performed as previously described [34]. In brief, A549 cells were seeded at 3 × 10^5^ cells per well in a 12-well plate for infections. Calu-3 cells were seeded similarly onto rat tail collagen type I-coated plates (Corning, Kennebunk, ME, USA, catalog no. 356500). Cells were washed once with phosphate-buffered saline (PBS) before being infected with virus diluted in serum-free medium—RPMI for A549 cells or DMEM for Calu-3 cells. Virus (MOI = 5 PFU/cell) was absorbed for 1 h at 37 °C before the cells were washed 3 times with PBS and the medium was replaced with 2% FBS RPMI (A549 cells) or 4% FBS DMEM (Calu-3 cells). At the indicated timepoints, 200 mL of medium was collected to quantify the released virus by plaque assay and stored at −80 °C. For HCoV-OC43 infections, similar infection conditions and media were used; however, the virus was absorbed, and the infections were incubated at 33 °C rather than 37 °C.

Plaque assays were performed using VeroE6 cells for SARS-CoV-2 and HCoV-OC43 and VeroCCL81 cells for MERS-CoV. SARS-CoV-2 and MERS-CoV plaque assays were performed in 12-well plates at 37 °C. HCoV-OC43 plaque assays were performed in 6-well plates at 33 °C. In all cases, the virus was absorbed into cells for 1 h at the indicated temperatures before overlay was added. A liquid overlay was used (DMEM with 2% FBS, 1x sodium pyruvate, and 0.1% agarose). Cell monolayers were fixed with 4% paraformaldehyde and stained with 1% crystal violet after the following incubation times: SARS-CoV-2 and MERS-CoV, 3 days; HCoV-OC43, 5 days. All plaque assays were performed in biological triplicate and technical duplicate.

**Pharmacologic agents**. Tunicamycin (Sigma-Aldrich, St. Louis, MO, USA, catalog no. T7765) and thapsigargin (Sigma-Aldrich catalog no. T9033) were purchased at >98% purity. For use in tissue culture, tunicamycin and thapsigargin stock solutions were prepared by dissolving in sterile dimethyl sulfoxide (DMSO). Salubrinal (catalog no. HY-15486) and Sal003 (catalog no. HY-15969) were purchased from MedChemExpress (Monmouth Junction, NJ, USA) and stock solutions were prepared in DMSO. Both compounds were diluted to the desired concentration in media and filter-sterilized before use in cell culture. The CC_50_ for salubrinal was determined at 48 h post-treatment using Cell Titer-Glo 2.0 (Promega, Madison, WI, USA, catalog no. G9242) according to the manufacturer’s protocol. The EC_50_ of salubrinal against HCoV-OC43 was determined in A549^ACE2^ cells. A549^ACE2^ cells were infected at an MOI of 0.1 for 1 h at 37 °C. The cells were then washed 3 times with PBS, and the indicated concentration of salubrinal in 2% FBS RPMI was added. At 48 h post infection, viral titers were determined via plaque assay. GraphPad Prism was used to calculate CC_50_ and EC_50_ using non-linear regression analysis—[Inhibitor] vs. response—variable slope (four parameters).

**Immunoblotting**. Cells were washed once with ice-cold PBS, and lysates were harvested at the indicated times post infection with lysis buffer (1% NP-40, 2 mM EDTA, 10% glycerol, 150 mM NaCl, 50 mM Tris HCl, pH 8.0) supplemented with protease inhibitors (Roche complete Mini—Millipore Sigma, St. Louis, MO, USA, catalog no. 11836170001) and phosphatase inhibitors (Roche PhosStop–Millipore Sigma catalog no. 4906845001). After 5 min, lysates were collected and mixed 3:1 with 4x Laemmli sample buffer (Bio-Rad, Hercules, CA, USA, catalog no. 1610747). Samples were heated at 95 °C for 10 min and then separated on SDS-PAGE and transferred to polyvinylidene difluoride (PVDF) membranes. Blots were blocked with 5% nonfat milk and probed with antibodies (Appendix A) diluted in the same blocking buffer. Primary antibodies were incubated overnight at 4 °C or for 1 h at room temperature. All secondary antibody incubation steps were carried out for 1 h at room temperature. Blots were visualized using Thermo Scientific SuperSignal chemiluminescent substrates (catalog no. 34095 or 34080).

**PhosTag Immunoblotting**. 7% acrylamide gels were poured containing 50 μM Phosbind acrylamide (ApexBio, Houston, TX, USA, catalog no. F4002) and 100 μM Mn^2+^. Equal volumes of samples were loaded into each well and run alongside an EDTA-free protein marker (ApexBio catalog no. F4005) at 100 V for approximately 3 h. Gels were washed 3 times in transfer buffer with 10% methanol and 10 mM EDTA for 20 min each. Three more washes of 10 min each with transfer buffer not containing EDTA were then performed. Transfers were performed as above with a 10% methanol transfer buffer. Proteins were imaged as above using the PERK antibody indicated in Appendix A.

**RNA sequencing**. Raw FastQ files were obtained from the Gene Expression Omnibus database (GSE193169). Read quality was assessed using FastQC v0.11.2 [35]. Raw sequencing reads from each sample were quality- and adapter-trimmed using BBDuk 38.73 [36]. The reads were mapped to the human genome (hg38 with Ensembl v98 annotation) using Salmon v0.13.1 [37]. Differential expression between mock, 36 hpi, and 48 hpi experimental conditions was analyzed using the raw gene counts files by DESeq2 v1.22.1 [38]. Volcano plots were generated using EnhancedVolcano v1.14.0 [39].

**Gene set enrichment analyses**. Gene set enrichment analysis (GSEA) was used to identify the upregulation of cellular pathways and responses. fgsea v1.22.0 [40] was used to perform specific gene set enrichment analyses and calculate normalized enrichment score (NES) and p-adjusted values on each dataset using DESeq2 stat values. Specific enrichment plots for the Reactome Unfolded Protein Response gene set (stable identifier R-HSA-381119; https://www.gsea-msigdb.org/gsea/msigdb/human/geneset/REACTOME_UNFOLDED_PROTEIN_RESPONSE_UPR.html; accessed on 5 August 2023) were generated using fgsea.

**Statistical Analysis**. All statistical analyses and plotting of data were performed using GraphPad Prism software (v9.5.1). RT-qPCR data were analyzed by one-way analysis of variance (ANOVA). Plaque assay data were analyzed by two-way ANOVA with multiple-comparison correction. Displayed significance is determined by the *p* value; *, *p* < 0.05; **, *p* < 0.01; ***, *p* < 0.001; ****, *p* < 0.0001; ns, not significant.

**Quantitative PCR (RT-qPCR)**. Cells were lysed with RLT Plus buffer, and total RNA was extracted using the RNeasy Plus minikit (Qiagen, Germantown, MD, USA, catalog no. 74134). RNA was reverse-transcribed into cDNA with a high-capacity cDNA reverse transcriptase kit (Applied Biosystems catalog no. 4387406). cDNA samples were diluted in molecular biology-grade water and amplified using specific RT-qPCR primers (Appendix A). RT-qPCR experiments were performed on a QuantStudio 3 PCR system (Thermo Fisher, Waltham, MA, USA) instrument. iQ SYBR Green Supermix was from Bio-Rad (catalog no. 1708880). Host gene expression displayed as the fold change over mock-infected samples was generated by first normalizing cycle threshold (C_T_) values to 18S rRNA to generate ΔC_T_ values (ΔC_T_ = C_T_ gene of interest—C_T_ 18S rRNA). Next, Δ(ΔC_T_) values were determined by subtracting the mock-infected ΔC_T_ values from the virus-infected samples. Technical triplicates were averaged and means were displayed using the equation 2^−Δ(ΔCt)^. Primer sequences are listed in Appendix A.

## 3. Results

### 3.1. HCoV-OC43, SARS-CoV-2, and MERS-CoV Activate the Unfolded Protein Response

To understand how different betacoronaviruses interact with the host, we analyzed previously published [34] transcriptomic RNA sequencing (RNA-seq) data from infected A549 lung cell lines expressing either dipeptidyl peptidase 4 (A549^DPP4^) to facilitate MERS-CoV infection or angiotensin-converting enzyme 2 (A549^ACE2^) for SARS-CoV-2 infection and HCoV-OC43. In all infections (MOI = 1 PFU/cell), we observed a distinct upregulation of the unfolded protein response (UPR) genes, especially PERK-regulated genes. Volcano plots were generated for each dataset, with selected UPR-regulated genes for MERS-CoV (Figure 2A), SARS-CoV-2 (Figure 2B), and HCoV-OC43 (Figure 2C) highlighted in red. HCoV-OC43 infection significantly upregulated the largest number of UPR-related genes (Figure 2C) compared to SARS-CoV-2 (Figure 2B) or MERS-CoV (Figure 2A). However, all three viruses strongly upregulated three PERK/ISR-regulated genes (labeled in Figure 2): *ATF3* [41]; DNA damage-inducible transcription factor 3 (*DDIT3)*, encoding CHOP; and *PPP1R15A,* encoding GADD34 [1]. As we recently reported, SARS-CoV-2 failed to induce IRE1α-regulated genes (Figure 2B) while MERS-CoV and HCoV-OC43 did [34]. Gene set enrichment analysis (GSEA) also showed significant upregulation of UPR-related genes during MERS-CoV (Figure 2D) and HCoV-OC43 (Figure 2F) infection, while SARS-CoV-2 (Figure 2E) did not significantly enrich this pathway.

### 3.2. MERS-CoV and HCoV-OC43 Do Not Induce p-eIF2α Despite PERK Activation

To confirm that PERK is activated during infection by these betacoronaviruses, A549 cells expressing the appropriate viral receptor were infected at a multiplicity of infection (MOI) of 5 PFU/cell. In addition to SARS-CoV-2, HCoV-OC43, and MERS-CoV, we also included MERS-CoV-nsp15^mut^/ΔNS4a, an immunostimulatory double mutant encoding a catalytically inactive nsp15 endoribonuclease and a deletion of the NS4a encoded protein [20]. Whole-cell lysates were collected at 24, 48, and 72 h post-infection (hpi) for immunoblot analysis. Due to the lack of effective phospho-PERK antibodies for human samples, PERK activation was assessed using Phos-tag^TM^ SDS-PAGE, which slows the migration of phosphorylated proteins through the polyacrylamide, thus separating phosphorylated and unphosphorylated species. As positive controls, cells were treated with thapsigargin (Tg), a sarcoendoplasmic reticulum calcium ATPase (SERCA) inhibitor [42], for one hour, or tunicamycin (TM), an N-linked glycosylation inhibitor [34], for eight hours to induce ER stress. These treatments induced PERK phosphorylation as evidenced by visualization of separated PERK and p-PERK bands (Figure 3A–C). Lysates collected from cells infected with each virus showed an upper band in these blots representing p-PERK, demonstrating PERK activation during infection, with nearly all the PERK being phosphorylated during HCoV-OC43 infection. PERK activation can also be visualized by standard SDS-PAGE, with virus-infected cells or cells treated with either Tg or TM. A band shift and shading pattern is observed, indicating PERK phosphorylation and activation (Figure 3D–F). This led us to conclude that all three viruses activate PERK during infection.

As we previously reported, wild-type (WT) MERS-CoV fails to induce PKR activation (indicated by PKR phosphorylation) or eIF2α phosphorylation up to 72 hpi (Figure 3D) [18]. In contrast, MERS-CoV-nsp15^mut^/ΔNS4a, which induces increased levels of dsRNA, strongly induced p-PKR and p-eIF2α [20], confirming that parental MERS-CoV effectively antagonizes PKR to limit eIF2α phosphorylation. Similar to WT MERS-CoV, HCoV-OC43 (Figure 3F) also failed to activate PKR or induce p-eIF2α during infection, although the mechanism of PKR antagonism remains unclear. However, SARS-CoV-2 robustly activated PKR and induced p-eIF2α over the course of infection (Figure 3E) [18].

It is striking that, despite the activation of at least one ISR kinase during infection and apparent ISR gene induction, WT MERS-CoV (Figure 3D) and HCoV-OC43 (Figure 3F) still fail to induce p-eIF2α during infection. To further assess ISR activation we next examined ATF4 translation during infection, which should occur rapidly following eIF2α phosphorylation [7]. As expected, ATF4 is readily detectable in cells treated with either Tg or TM. However, during infection with any of the three viruses, with or without the presence of p-eIF2α, ATF4 could not be detected at any timepoint (Figure 3D–F). This has been reported previously by other groups probing for ATF4 during infections with coronaviruses [43,44]; however, it is still unclear why this occurs. Despite the absence of detectable ATF4 during infection with any virus, ATF4-regulated genes were highly upregulated. MERS-CoV (Figure 3G) and HCoV-OC43 (Figure 3I) both induced ATF3, GADD34, and CHOP at increasing levels over the course of infection. While HCoV-OC43 induced much higher levels of GADD34 compared to MERS-CoV, CHOP induction by MERS-CoV dwarfed the other viruses, matching recent reports that MERS-CoV strongly induces apoptosis through PERK and CHOP signaling [45,46]. Interestingly, SARS-CoV-2 (Figure 3H) also induced ATF3 and GADD34 throughout the course of infection but failed to significantly upregulate CHOP. This indicates that, while PERK activation and signaling is a common feature of betacoronavirus infection, there are differences in the induction of certain responses that remain to be explored.

To understand the absence of eIF2α phosphorylation despite PERK activation during MERS-CoV and HCoV-OC43 infection, we probed for GADD34 protein expression. GADD34 was translated following Tg or TM treatment, confirming that this pathway can be induced in as little as 1 h following ER stress. Consistent with the transcriptional induction of GADD34 (Figure 3G–I), GADD34 protein expression was also observed over the course of MERS-CoV, SARS-CoV-2, and HCoV-OC43 infection (Figure 3D–F). This suggested that GADD34 expression during WT MERS-CoV and HCoV-OC43 infection may be keeping p-eIF2α levels below the limit of detection for immunoblotting. However, SARS-CoV-2 does not seem to share this characteristic, perhaps due to the combined activity of PERK (Figure 3B) and PKR (Figure 3E), or via another mechanism. The ability of cells to dephosphorylate eIF2α during TM treatment has been noted in the literature [47] and demonstrates that GADD34 is capable of promoting dephosphorylation of eIF2α despite continued ER stress.

### 3.3. Betacoronaviruses Promote Translational Shutoff with or Without p-eIF2α

To understand the impact on overall translation in cells infected with each betacoronavirus, we utilized puromycin incorporation to visualize nascent peptide production. Cells were infected with each virus (MOI = 5 PFU/cell), and at the indicated timepoints, puromycin was added to the media for incorporation into nascent peptide chains. Whole-cell lysates were then collected, subjected to immunoblotting, and stained with an antibody raised against puromycin as a measure of total protein translation and with viral nucleocapsid (N) antibody, which served as a marker of infection and a readout of viral protein synthesis [19]. Tg treatment served as a positive control for ER stress and p-eIF2α-mediated translational attenuation (Figure 4C).

Figure 4A shows infection with WT MERS-CoV or the MERS-CoV nsp15^mut^/ΔNS4a double mutant virus that induces p-eIF2α during infection [20] (see Figure 3D). Immunoblots for puromycin incorporation revealed that WT MERS-CoV produces a progressive shutdown of host translation despite the lack of p-eIF2α during infection, while conversely, viral translation of N increased over the course of infection. MERS-CoV- nsp15^mut^/ΔNS4a, which activates PKR and induces p-eIF2α during infection, promotes a faster translational shutoff by 24 hpi, supporting the role of p-eIF2α in limiting translation during CoV infection. However, both viruses appear to reach similar levels of translational attenuation at 48 h post infection. In contrast to the progressive translational shutoff induced by WT MERS-CoV infection, SARS-CoV-2 appears to rapidly reduce host translation to very low levels within 24 h of infection, with puromycin incorporation remaining low at all timepoints examined (Figure 4B). However, SARS-CoV-2 N, similar to MERS-CoV N, continues to be translated despite the very low levels of global translation within infected cells. HCoV-OC43 infection also induced a rapid shutoff of translation within infected cells that was similar to the attenuation induced by Tg treatment (Figure 4C). This was surprising because HCoV-OC43, like WT MERS-CoV, fails to induce p-eIF2α during infection.

### 3.4. eIF2α Dephosphorylation Is a Druggable Target During Betacoronavirus Infection

Since WT MERS-CoV and HCoV-OC43 both limit eIF2α phosphorylation during infection and p-eIF2α is detrimental to MERS-CoV infection [20], we explored if the inhibition of GADD34 during betacoronavirus infection would limit viral replication. GADD34 has been reported to be inhibited by several compounds that target the GADD34:PP1 holoenzyme [48]. Of these, salubrinal [32] has been utilized widely in the literature. Therefore, salubrinal was used during infection to test its therapeutic potential against betacoronaviruses.

We began by assessing the effects of salubrinal on HCoV-OC43 and SARS-CoV-2 infection because these two viruses induced different eIF2α phenotypes while being able to replicate within the same A549^ACE2^ cell line. Cytotoxicity of salubrinal was determined in A549^ACE2^ cells after 48 h of treatment. We determined that salubrinal is not toxic at concentrations below 160 μM (CC_50_ ≥ 160 μM) (Appendix A). Additionally, the EC_50_ for HCoV-OC43 was determined to be 2.58 μM (Appendix A). However, a sharp decline in HCoV-OC43 titers was observed at 20µM salubrinal (Appendix A). This matches with literature reporting the approximate IC_50_ value of salubrinal for inhibiting GADD34 in cells to be 15µM. Based on these data and the common use of 20 µM of salubrinal in the literature, [49,50], we used this dose in our experiments. Thus, cells were mock-infected or infected with HCoV-OC43 or SARS-CoV-2 (MOI= 5 PFU/cell) and incubated for 24 h to establish viral infection before 20 µM salubrinal or Sal003 [49], a salubrinal derivative with a similar function, was added for 4 or 24 h. Whole-cell lysates were collected and analyzed by immunoblot (Figure 5A,B). HCoV-OC43 and SARS-CoV-2 activated PERK and induced GADD34 expression with or without inhibitor treatment. However, during HCoV-OC43 infection, salubrinal or Sal003 treatment was required to induce p-eIF2α, confirming that this inhibitor can promote p-eIF2α (Figure 5A). In contrast, immunoblots of SARS-CoV-2-infected cells demonstrated no difference in p-eIF2α induction, with or without drug treatment (Figure 5B).

We next examined the impact on viral replication when A549^ACE2^ cells were treated with 20 μM of salubrinal immediately after infection (MOI = 0.1 PFU/cell). HCoV-OC43 showed high sensitivity to salubrinal, with infectious virus titers measured by plaque assay being reduced by approximately 10-fold at 24 hpi with salubrinal treatment and 100-fold at 48 hpi and 72 hpi (Figure 5C). In contrast, SARS-CoV-2 infections demonstrated no defect in viral replication (Figure 5C). Similar treatments in A549^DPP4^ cells infected with MERS-CoV or MERS-CoV nsp15^mut^/ΔNS4a were performed (Appendix A). Examining MERS-CoV replication, MERS-CoV-nsp15^mut^/ΔNS4a is attenuated [20], displaying 2- to 5-fold fewer PFU/mL released at each timepoint compared to the WT virus (Appendix A). Salubrinal treatment reduced WT MERS-CoV and MERS-CoV nsp15^mut^/ΔNS4a titers by 10- to 100-fold at each timepoint examined in A549^DPP4^ cells. Overall, these data demonstrate that replication of MERS-CoV and HCoV-OC43 is sensitive to salubrinal treatment and inhibition of eIF2α dephosphorylation during infection, while SARS-CoV-2 is not.

### 3.5. GADD34 Knockout Only Slightly Impacts HCoV-OC43 Replication

To validate our results using salubrinal we utilized CRISPR-Cas9 in our A549^ACE2^ cells to knock out GADD34 or introduced a control, scrambled single guide RNA (sgRNA). GADD34 knockout (KO) was validated using GADD34 expression by immunoblot and by assessing translational recovery during Tg treatment (Appendix A). As observed in Appendix A, control sgCtrl cells produce GADD34 protein and begin to recover translation after only two hours of Tg treatment, with levels of translation steadily increasing over four hours. In contrast, GADD34-KO cells fail to produce GADD34 protein or restart translation at any point (Appendix A), confirming the loss of GADD34. Two separate GADD34-KO clones (clone 15 and clone 23) were chosen for infection with either HCoV-OC43 or SARS-CoV-2.

The sgCtrl generated clone and both GADD34-KO clones were infected with SARS-CoV-2 or HCoV-OC43 (MOI = 2 PFU/cell). Infected cells showed robust PERK activation, as assessed by immunoblot analysis of whole-cell lysates harvested from cells following treatment with Tg or infection with HCoV-OC43 (Figure 6A) or SARS-CoV-2 (Figure 6C). Phosphorylation of eIF2α was also detected following Tg treatment and over the course of SARS-CoV-2 infection (Figure 6C). No consistent impact on SARS-CoV-2 infectious virus production was observed over the time course (Figure 6D). However, p-eIF2α was not induced during HCoV-OC43 infection of sgCtrl cells nor in infections of both GADD34-KO clones (Figure 6A). Thus, GADD34 KO does not appear to significantly alter the phosphorylation state of eIF2α during HCoV-OC43 or SARS-CoV-2 infection. Loss of GADD34 also failed to consistently reduce HCoV-OC43 titers in either knockout clone, with the single small but statistically significant difference at 72 hpi most likely not being a biologically significant reduction (Figure 6B). This suggests that our hypothesis regarding the role of GADD34 in HCoV-OC43 infection is incorrect.

It is surprising that GADD34 KO is not as effective as salubrinal, a known GADD34 inhibitor, at reducing HCoV-OC43 titers. While salubrinal has been reported to inhibit PP1:GADD34 [32,51,52], it has also been reported to inhibit the PP1 holoenzyme in complex with CReP [32]. Thus, the additional efficacy of salubrinal may be due to the co-inhibition of CReP during HCoV-OC43 infection. We investigated CReP expression at the RNA level by RT-qPCR and at the protein level by immunoblotting of lysates from cells infected with HCoV-OC43 or SARS-CoV-2. Surprisingly, we observed a dramatic increase in CReP mRNA levels during HCoV-OC43 infection (3-8-fold) (Figure 6E) as well as a dramatic increase in CReP protein levels (Figure 6A). Conversely, SARS-CoV-2 infection reduced CReP expression at the RNA and protein expression levels during infection (Figure 6E,D).

### 3.6. CReP Is Necessary for Efficient HCoV-OC43 Replication

To investigate the role of CReP in betacoronavirus replication, we utilized siRNA to knock down (KD) CReP expression in A549^ACE2^ cells before infecting with HCoV-OC43 or SARS-CoV-2. CReP protein levels were efficiently reduced compared to treatment with a scrambled siRNA control (siCtrl) (Figure 7A,C). Seventy-two hours after siRNA transfection, cells were infected with HCoV-OC43 or SARS-CoV-2 (MOI = 2 PFU/cell) for the indicated times, and whole-cell lysates were collected and analyzed by immunoblotting. During the infection of siCtrl-treated cells with HCoV-OC43, we observed a decrease in p-eIF2α levels below the background of mock-infected siCtrl cells. Knockdown of CReP was maintained through the course of infection and led to an increase in p-eIF2α levels, particularly at 24 hpi (Figure 7A). This increase in p-eIF2α at 24 hpi also corresponded with a notable decrease in HCoV-OC43 N protein. CReP KD also reduced HCoV-OC43 titers by approximately 100-fold at 24 hpi, with the defect decreasing to only 10-fold at 48 hpi and 3-fold at 72 hpi (Figure 7B). We hypothesize that this diminishing effect on viral replication as the infection progresses may be due to CReP upregulation paired with siRNA turnover. These data, as well as the significant impact of salubrinal treatment on HCoV-OC43 replication (Figure 5C), leads us to conclude that HCoV-OC43 preferentially promotes eIF2α dephosphorylation and viral replication through CReP rather than GADD34.

In contrast to HCoV-OC43 infection, CReP KD during SARS-CoV-2 infection failed to have a major impact on p-eIF2α levels. Due to cell death at the MOI used, a small decrease in p-eIF2α levels at 48 hpi with both CReP KD and siCrtl was observed (Figure 7C). This KD of CReP failed to reduce SARS-CoV-2 replication (Figure 7D), supporting the ability of SARS-CoV-2 to circumvent cellular translational control.

### 3.7. CReP and GADD34 Both Contribute to HCoV-OC43 Replication

Having examined the individual contributions of GADD34 and CReP to betacoronavirus replication, we next combined these conditions to determine if CReP KD and GADD34 KO would have a combinatorial effect on HCoV-OC43 replication. To do this, we treated sgCtrl A549^ACE2^ cells or GADD34-KO cells (clone 23) with scrambled (siCtrl) or CReP-targeting (siCReP) siRNA. Immunoblots of whole-cell lysates harvested from HCoV-OC43-infected cells (MOI = 2 PFU/cell) at 24 hpi (Figure 8A) and 48 hpi (Figure 8B) were performed. As expected, GADD34 expression was ablated in GADD34-KO cells, while CReP expression was efficiently reduced with siRNA treatment in either cell line at both timepoints. As observed in Figure 7, CReP KD in either sgCtrl or GADD34-KO cells led to an increase in p-eIF2α during HCoV-OC43 infection at 24 hpi and 48 hpi (Figure 8A,B). Additionally, GADD34 KO alone did not lead to increased p-eIF2α phosphorylation levels (Figure 8A,B) and failed to impact HCoV-OC43 replication (Figure 8E). In contrast, CReP KD in sgCtrl cells significantly reduced HCoV-OC43 titers by nearly 50-fold at 24 hpi, with this difference again diminishing at later timepoints. However, combining CReP KD in GADD34-KO cells led to an even greater decrease in HCoV-OC43 titers, reducing viral replication another 5-fold compared to CReP KD alone (Figure 8E). This difference was sustained but again diminished by 48 and 72 hpi. Together, these data suggest that while CReP is the main driver for eIF2α dephosphorylation and HCoV-OC43 replication, GADD34 also plays a role in boosting viral replication.

As expected, neither CReP KD, GADD34 KO, nor the combination of the two significantly altered the induction of p-eIF2α during SARS-CoV-2 infection (MOI = 2 PFU/cell) at 24 hpi (Figure 8C) or 48 hpi (Figure 8D). Additionally, despite the loss of GADD34, the reduction in CReP, or a combination of the two, SARS-CoV-2 replication was again unchanged under any condition tested (Figure 8F).

## 4. Discussion

We have presented evidence that HCoV-OC43, SARS-CoV-2, and MERS-CoV—representing different betacoronavirus subgenera [31]—activate the PERK arm of the ISR/UPR. The analysis of RNA-seq data from infections of A549 cells with each virus [34] demonstrated enrichment of ISR-regulated genes, including *ATF3* [41], GADD34 (gene name *PPP1R15A*), and CHOP (gene name *DDIT3*) (Figure 2) [1]. We have previously shown that MERS-CoV effectively antagonizes PKR during infection and fails to induce phosphorylation of eIF2α, while SARS-CoV-2 infection activates PKR and induces p-eIF2α [18]. However, we have also shown that cells lacking PKR still phosphorylate eIF2α during SARS-CoV-2 infection, suggesting that at least one other ISR kinase is active [18]. Due to the remodeling of the host’s ER during coronavirus infection [14] and observations that overexpression of spike protein alone is sufficient to induce the UPR [43,53], we hypothesized that PERK activation during infection with these viruses was contributing to the responses observed in our RNA-seq data.

Despite confirming PERK activation and downstream signaling during CoV infection, we observed that WT MERS-CoV and HCoV-OC43 failed to induce detectable p-eIF2α (Figure 3). Having shown the induction of GADD34 during infection with each virus, the most parsimonious explanation for this disconnect is that GADD34 is driving eIF2α dephosphorylation [13] during WT MERS-CoV and HCoV-OC43 infection. Indeed, our positive controls Tg and TM reveal this process in action in A549 cells. As shown in Figure 3D–F, 1 h of Tg treatment is sufficient to activate PERK, induce p-eIF2α, and promote ATF4 and GADD34 translation. Eight hours of TM treatment similarly induces PERK activation and ATF4/GADD34 translation. However, at this timepoint, there is no longer detectable p-eIF2α, because enough GADD34 has accumulated to dephosphorylate eIF2α. Such instances of viruses preferring the dephosphorylated state of eIF2α have been observed with pseudorabies virus, and characterization of viral proteins with similar functions to GADD34 demonstrate their need to maintain translation during infection [54,55,56]. However, we and others have shown that coronaviruses mediate host translational shutdown during infection using non-structural protein (nsp)1 [57,58,59,60,61], even without the induction of p-eIF2α. Despite this, p-eIF2α is detrimental to MERS-CoV replication and protein production [20], highlighting the struggle for translational control between virus and host during infection. It is thus intriguing that SARS-CoV-2 shows efficient N production despite continuous phosphorylation of eIF2α during infection (Figure 4B). This suggests that MERS-CoV and HCoV-OC43, but not SARS-CoV-2, require a specific translational context within the infected cell to replicate optimally.

To investigate the impact of eIF2α dephosphorylation on betacoronavirus infection, we utilized salubrinal, a widely used inhibitor of eIF2α dephosphorylation. This compound has been reported to target the PP1:GADD34 and PP1:CReP holoenzymes to disrupt eIF2α dephosphorylation [32,48], thus making it a potential host-directed antiviral for coronavirus infection. We found that salubrinal treatment of A549 cells is effective against HCoV-OC43 (Figure 5) and MERS-CoV (Appendix A) replication. However, SARS-CoV-2 showed little, if any, sensitivity to salubrinal treatment (Figure 5B,C). It is unclear what could be mediating this difference, and more research will be required to uncover the exact mechanism. It is also notable that the extreme sensitivity of HCoV-OC43 to salubrinal treatment may distinguish this common cold coronavirus from the lethal human coronaviruses.

Due to the nonspecific nature of small molecule inhibitors, we utilized a CRISPR-Cas9 KO of GADD34 to confirm its role in HCoV-OC43 and SARS-CoV-2 infection. Due to the similar phenotypes between HCoV-OC43 and MERS-CoV and the ability of HCoV-OC43 to infect the same A549^ACE2^ cell line as SARS-CoV-2, we proceeded to compare only HCoV-OC43 and SARS-CoV-2. In contrast to our initial hypothesis, GADD34-KO cells showed no detectable alterations in p-eIF2α levels (Figure 6A,C) or viral replication (Figure 6B,D) during HCoV-OC43 or SARS-CoV-2 infection. These results are supported by similar findings that were recently published [62], although we have further expanded upon this to provide a potential explanation for our shared negative results. A dramatic increase in CReP mRNA and protein levels was observed during HCoV-OC43 infection (Figure 6A,E), while a reduction of both was seen during SARS-CoV-2 infection (Figure 6E,C). Thus, our data suggest that CReP, another target of salubrinal [32], is the main driver of eIF2α dephosphorylation during HCoV-OC43 infection.

Supporting the role of CReP in dephosphorylation of eIF2α, we found that knocking down CReP expression using siRNA led to increased p-eIF2α levels, decreased N expression (Figure 7A), and a significant reduction in viral titers (Figure 7B) during HCoV-OC43 infection. SARS-CoV-2 replication (Figure 7D) and p-eIF2α levels (Figure 7C) once again remained unchanged. To understand if GADD34 and CReP are working cooperatively during HCoV-OC43 infection, CReP KD in GADD34-KO cells was performed. These data clearly show a combinatorial role for these PP1 binding partners during HCoV-OC43 infection due to CReP KD in GADD34-KO cells having a more dramatic effect on HCoV-OC43 replication than CReP KD alone (Figure 8E). Thus, we conclude that CReP is the primary factor for promoting dephosphorylation of eIF2α during HCoV-OC43 infection, but that GADD34 also plays a role in optimizing HCoV-OC43 replication. In contrast to this, SARS-CoV-2 was still unaffected by the combined loss of GADD34 and CReP (Figure 8F), and p-eIF2α levels were unaltered during infection of any condition (Figure 8C,D).

We thus conclude that HCoV-OC43 and SARS-CoV-2 have diverged in their tolerance to host translational control via eIF2α phosphorylation. HCoV-OC43 appears to employ multiple mechanisms to limit eIF2α phosphorylation, including antagonizing PKR (Figure 3F), upregulating GADD34 (Figure 3I) and CReP (Figure 6E), and promoting eIF2α dephosphorylation (Figure 8A,B). In contrast, SARS-CoV-2 diverges from HCoV-OC43 in all of these aspects and promotes sustained eIF2α phosphorylation throughout the course of infection (Figure 3E), limited GADD34 upregulation (Figure 3H), and decreased CReP expression (Figure 6E). We hypothesize that SARS-CoV-2 may benefit from eIF2α phosphorylation and, thus, may both induce phosphorylation and limit dephosphorylation to maximize cellular translational shutoff. How SARS-CoV-2 can escape the deleterious effects of p-eIF2α while other betacoronaviruses cannot remains to be determined. It is possible that SARS-CoV-2 has evolved a way to promote localized dephosphorylation of p-eIF2α around viral mRNAs [63], thus promoting even further skewing of cellular translation towards viral transcripts. Additionally, nsp1, the viral replicase protein that interacts with host ribosomes and promotes the selective translation of viral mRNAs [57], could play a role. Indeed, a recent study found that SARS-CoV-2 nsp1 binds to the initiation factors EIF1 and EIF1A to enhance the translation of viral transcripts [64]. Mechanisms such as this, as well as other undiscovered functions of SARS-CoV-2 replicase and accessory proteins, could help to maintain viral translation rates high under conditions of a translationally limited host.

It is surprising and unorthodox that CReP, which promotes continuous, low-level dephosphorylation, could compensate for the loss of GADD34 during intense ER stress, such as during coronavirus infection. However, studies that have suggested that CReP has a limited capability to compensate for GADD34 [10,63,65] did not include viral infection, which could alter typical function. For instance, during SARS-CoV-2 infection, we observed decreased CReP expression at the mRNA level and protein level (Figure 6C,E). Additionally, SARS-CoV-2 induced the lowest levels of GADD34 compared to HCoV-OC43 (compare Figure 3H,I) and MERS-CoV (compare Figure 3G,H). Thus, we conclude that HCoV-OC43 induces both GADD34 and CReP during infection, maximizing eIF2α dephosphorylation to maintain virus protein production. SARS-CoV-2, on the other hand, induces low levels of GADD34 and even decreases CReP levels, thus allowing continued eIF2α phosphorylation throughout infection while somehow not affecting SARS-CoV-2 protein production. MERS-CoV lies somewhere in the middle, relying on eIF2α dephosphorylation, but not to the same extent as HCoV-OC43. Targeting both GADD34 and CReP with salubrinal [32] may serve as an effective therapy against MERS-CoV and especially HCoV-OC43.

It remains unclear exactly how HCoV-OC43 and SARS-CoV-2 may be differentially regulating CReP expression during infection. Previous studies have reported that CReP can be negatively regulated by the IRE1α pathway of the unfolded protein response via regulated IRE1-dependent decay (RIDD), which degrades CReP mRNA [66]. However, we have previously reported that HCoV-OC43 strongly activates IRE1α during infection, while SARS-CoV-2 inhibits the activation of the IRE1α RNase domain [34]. This would be expected to produce the opposite regulation of CReP to what we observed during HCoV-OC43 and SARS-CoV-2 infection if RIDD were indeed involved (Figure 6E). CReP has also been found to be negatively regulated by mir-98-5p [67,68], which could be investigated in future studies during CoV infection. While the exact mechanism of CReP upregulation is unclear, it has been reported that CReP mRNA levels can increase to compensate for GADD34 loss under stress conditions, indicating that CReP expression might not always be constitutive [10]. We hypothesize that HCoV-OC43 induces such extreme levels of ER stress that this triggers the upregulation of not only GADD34 but also CReP as well. However, further studies will be necessary to unravel this connection.

While our findings regarding GADD34 and CReP during betacoronavirus infection are novel, other groups have reported on the role of PERK during MERS-CoV infection. These studies have concluded that MERS-CoV activates PERK during infection, leading to apoptosis through CHOP upregulation. Interestingly, they found that apoptosis mediated by PERK is beneficial to MERS-CoV replication, but not to SARS-CoV-2 [46], and PERK inhibitors are potentially antiviral to MERS-CoV [45]. This demonstrates that MERS-CoV must balance the negative impacts of PERK activation—eIF2α phosphorylation—to exploit this pathway, further supporting the potential efficacy of host-directed therapeutics. This further demonstrates that CoV interactions with the UPR are exceedingly complex and that there is much more to be explored regarding the PERK pathway and its intricate connections to translation, ER health, and cell fate.

Based on our findings, we propose eIF2α dephosphorylation as a potential host-directed therapeutic target during embeco- or merbecovirus infection. Salubrinal treatment led to reductions in MERS-CoV and HCoV-OC43 replication, while CReP depletion confirmed that this protein is necessary for optimal HCoV-OC43 replication and eIF2α dephosphorylation. Interestingly, HCoV-OC43 seems to require inhibition of both GADD34 and CReP to maximally reduce viral titers. Deletion of both GADD34 and CReP has been reported to be toxic to cells. In the case of GADD34 or CReP loss alone, the other can compensate and enable cell survival under conditions of stress [10]. Deletion of both prevents all eIF2α dephosphorylation and thus brings the ternary complex concentration to toxically low levels [51,65], which is why we could not produce a double knockout cell line, and limits the usefulness of long-term salubrinal treatment. Thus, single-target inhibitors such as Sephin1 [51] for GADD34 or Raphin1 [65] for CReP would be necessary for in vivo treatments, while limited doses or treatment courses of drugs such as salubrinal could be considered. Targeting ER stress has been proposed as a therapeutic strategy for coronaviruses before, with PERK inhibitors [45,46] (as discussed above), Tg [44], and TM analogs [69] being reported to be effective at combating coronavirus replication in cells. However, stress-inducing drugs are likely to have systemic toxicity [48] that, in cases of severe CoV infection, could harm already stressed organs. As viruses are much more sensitive to translational perturbations than their hosts [70,71,72], it is possible that rapid treatment with eIF2α dephosphorylation inhibitors could deliver a host-directed antiviral effect that primarily targets infected cells. However, our understanding of the interactions of coronaviruses with translation, eIF2α phosphorylation, and host cell stress responses is still in a very early stage, and much more investigation remains to be performed.

## Figures and Tables

**Figure 1 viruses-17-00120-f001:**
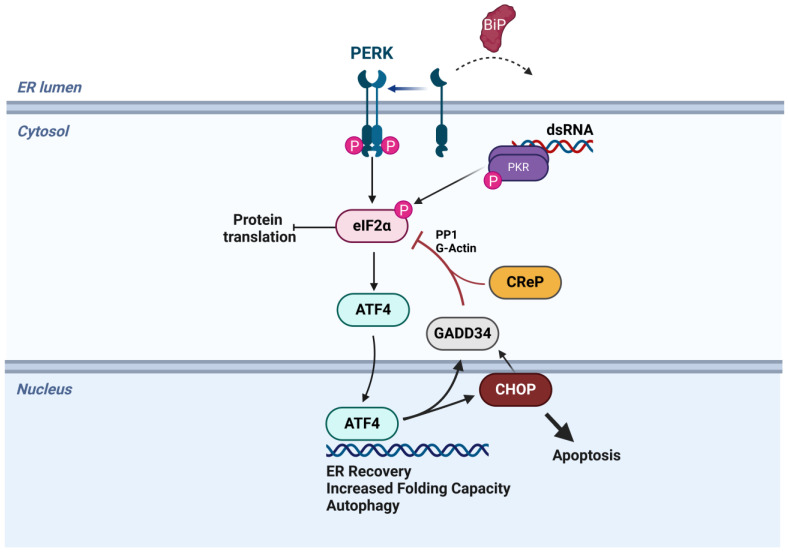
**Diagram of the PERK and PKR pathways from the integrated stress response.** Following activation of either PERK or PKR, serine 51 on eIF2α is phosphorylated, leading to translational attenuation and the upregulation of ATF4 translation. ATF4 induces a number of recovery responses. GADD34 and CReP promote eIF2α dephosphorylation to restart translation, and CHOP is a pro-apoptotic transcription factor that promotes death in terminally stressed cells. Created with BioRender.com.

**Figure 2 viruses-17-00120-f002:**
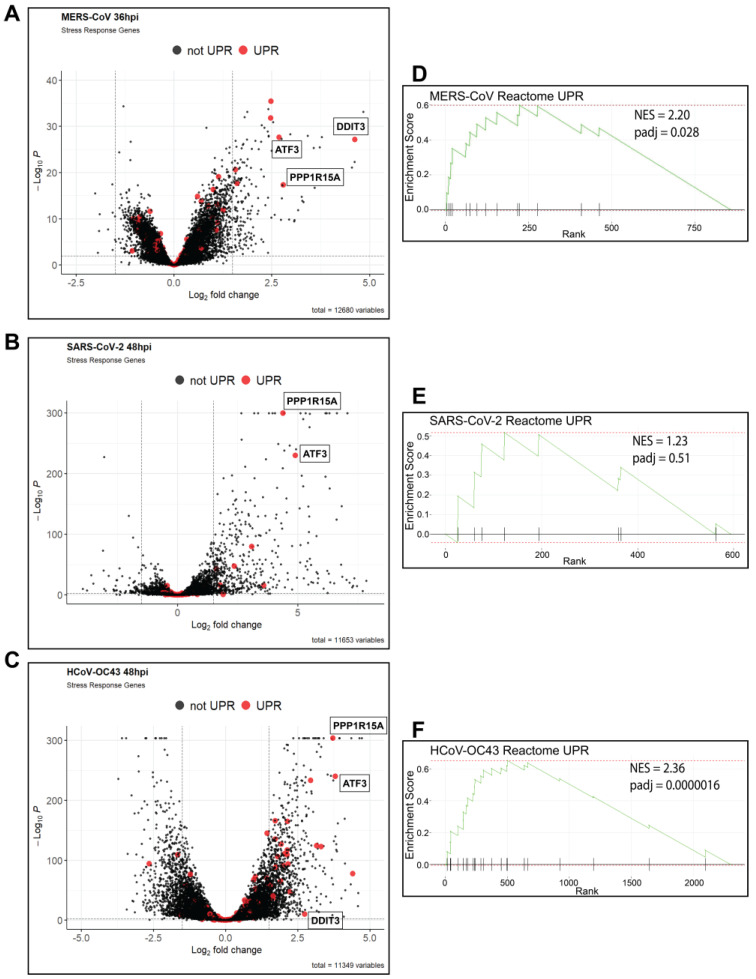
**MERS-CoV, SARS-CoV-2, and HCoV-OC43 display signatures of PERK and UPR activation.** (**A**–**C**) RNA-seq datasets of MERS-CoV infection in A549^DPP4^ cells at 36 hpi (**A**) and SARS-CoV-2 (**B**) and HCoV-OC43 (**C**) infection in A549^ACE2^ cells at 48 hpi (MOI = 1 PFU/cell) were compared to mock infections and differentially expressed genes called using DESeq2. UPR-regulated genes are-highlighted (in red). Volcano plots were generated using EnhancedVolcano. (**D**–**F**) Gene set enrichment analysis (GSEA) using the RNA-seq datasets from A-C. Pathway enrichment plots for the Reactome Unfolded Protein Response (UPR) gene list were generated for MERS-CoV (**D**), SARS-CoV-2 (**E**), and HCoV-OC43 (**F**) infected A549s. Normalized enrichment score (NES) and p-adjusted value (padj) are displayed on the plots.

**Figure 3 viruses-17-00120-f003:**
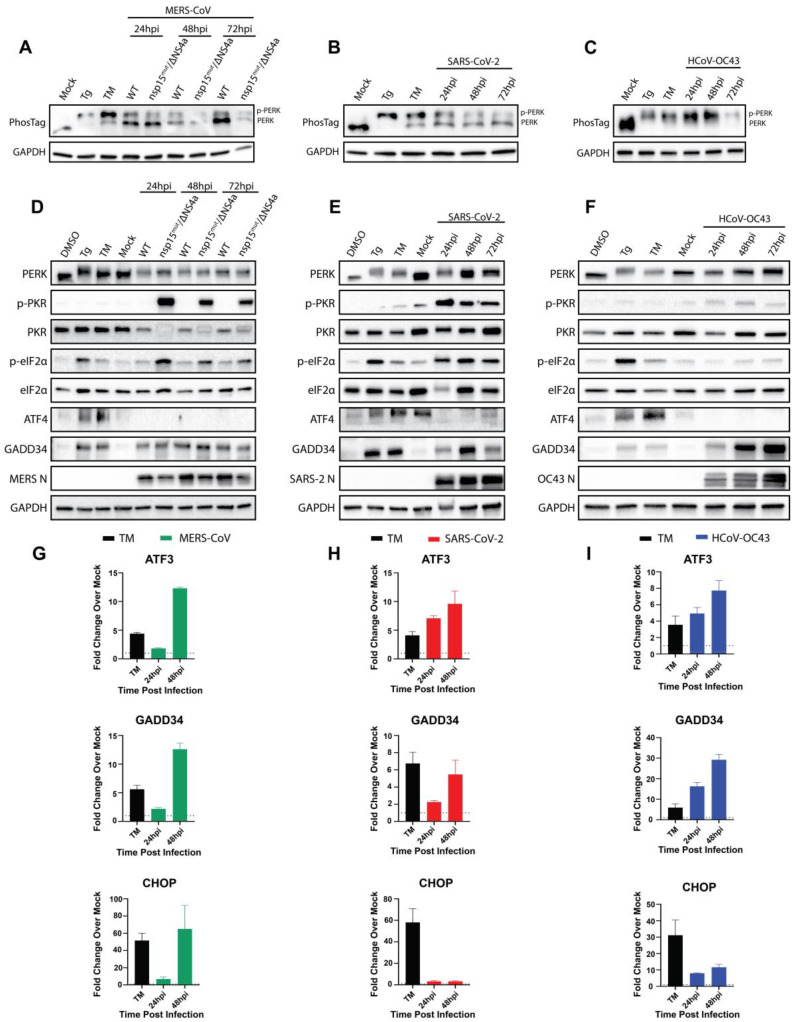
**MERS-CoV, SARS-CoV-2, and HCoV-OC43 all activate PERK and downstream signaling during infection.** In all blots (**A**–**F**), thapsigargin (Tg, 1 μM treatment for 1 h) and tunicamycin (TM, 1 μg/mL treatment for 8 h) served as positive controls, while DMSO (0.1%) served as a vehicle control. Cells (**A**—A549^DPP4^; **B**,**C**—A549^ACE2^) were infected with the indicated viruses (MOI = 5PFU/cell) or mock-infected, and whole-cell lysates were collected at the indicated timepoints. (**A**–**C**) Extracted proteins were resolved in SDS-polyacrylamide gels containing 50 μM Phosbind acrylamide and Mn^2+^ to separate phosphorylated and unphosphorylated proteins. Gels were transferred and immunoblotted for PERK (top gel—PhosTag). GAPDH run by standard SDS-PAGE served as a loading control. (**D**–**F**) Western immunoblots were performed by standard SDS-PAGE for the indicated proteins. (**G**–**I**) Cells were treated with DMSO or 1 μg/mL tunicamycin (TM) for 8 h before total RNA was extracted. (**G**) A549^DPP4^ cells were mock-infected or infected at MOI = 5 PFU/cell with MERS-CoV, and total RNA was extracted at the indicated timepoints. (**H**,**I**) A549^ACE2^ cells were mock-infected or infected with SARS-CoV-2 (**H**) or HCoV-OC43 (**I**) at MOI = 5 PFU/cell, and total RNA was collected at the indicated timepoints. Expression of the indicated genes was determined using RT-qPCR, with fold change over mock values being calculated as 2^−Δ(ΔCt)^.

**Figure 4 viruses-17-00120-f004:**
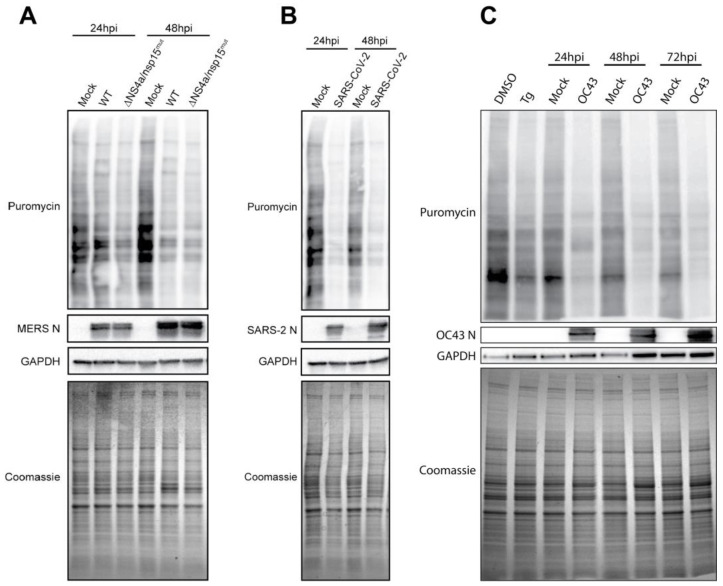
**Global translation during betacoronavirus infection.** A549 cells expressing the appropriate viral receptors were treated with 0.1% DMSO, 1 μM thapsigargin (Tg) for 1 h, mock-infected, or infected at MOI = 5 PFU/cell. At the indicated times, 10 μg/mL of puromycin was added to cells for 10 min before lysis and total protein collection. Samples were subjected to immunoblotting for the indicated proteins, while Coomassie staining was used as a readout of total protein. (**A**) MERS-CoV- or MERS-CoV nsp15^mut^/ΔNS4a-infected A549^DPP4^ cells. (**B**) SARS-CoV-2-infected A549^ACE2^ cells. (**C**) A549^ACE2^-infected HCoV-OC43 cells. N = nucleocapsid protein.

**Figure 5 viruses-17-00120-f005:**
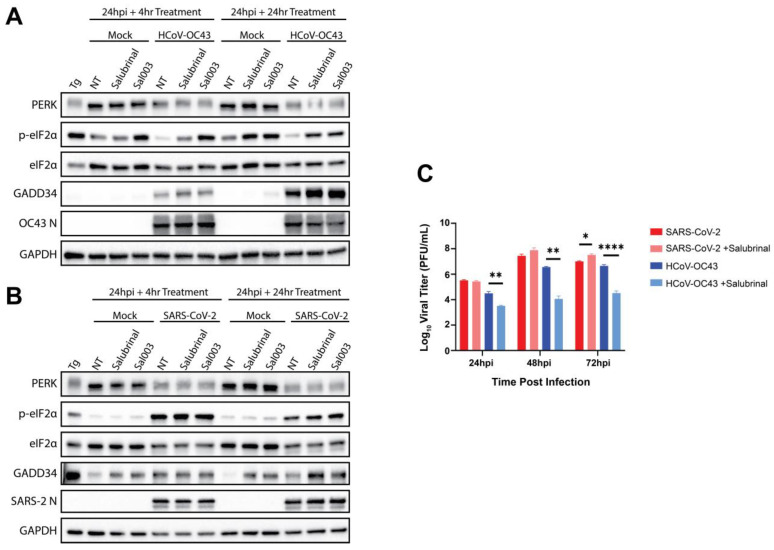
**Salubrinal treatment is effective against HCoV-OC43 but not SARS-CoV-2.** A549^ACE2^ cells were mock-infected or infected at MOI = 5 PFU/cell with HCoV-OC43 (**A**,**C**) or SARS-CoV-2 (**B**,**C**). (**A**,**B**) At 24 hpi, cell media was replaced with media containing 20 μM salubrinal or 20 μM Sal003, and infections were allowed to progress for 4 or 24 more hours. At the indicated timepoints, whole-cell lysates were collected. Immunoblotting was performed for the indicated proteins. NT = no treatment. Thapsigargin (Tg, 1 μM for 1 h) was used as a positive control for p-eIF2α. (**C**) A549^ACE2^ cells were infected with HCoV-OC43 or SARS-CoV-2 at MOI = 0.1 PFU/cell. Immediately following virus absorption, cells were treated with 20 μM salubrinal or 0.1% DMSO. At the indicated timepoints, supernatants were collected and infectious virus titers quantified by plaque assay. Statistics were calculated using 2-way ANOVA. * = *p* < 0.05; ** = *p* < 0.01; **** = *p* < 0.0001.

**Figure 6 viruses-17-00120-f006:**
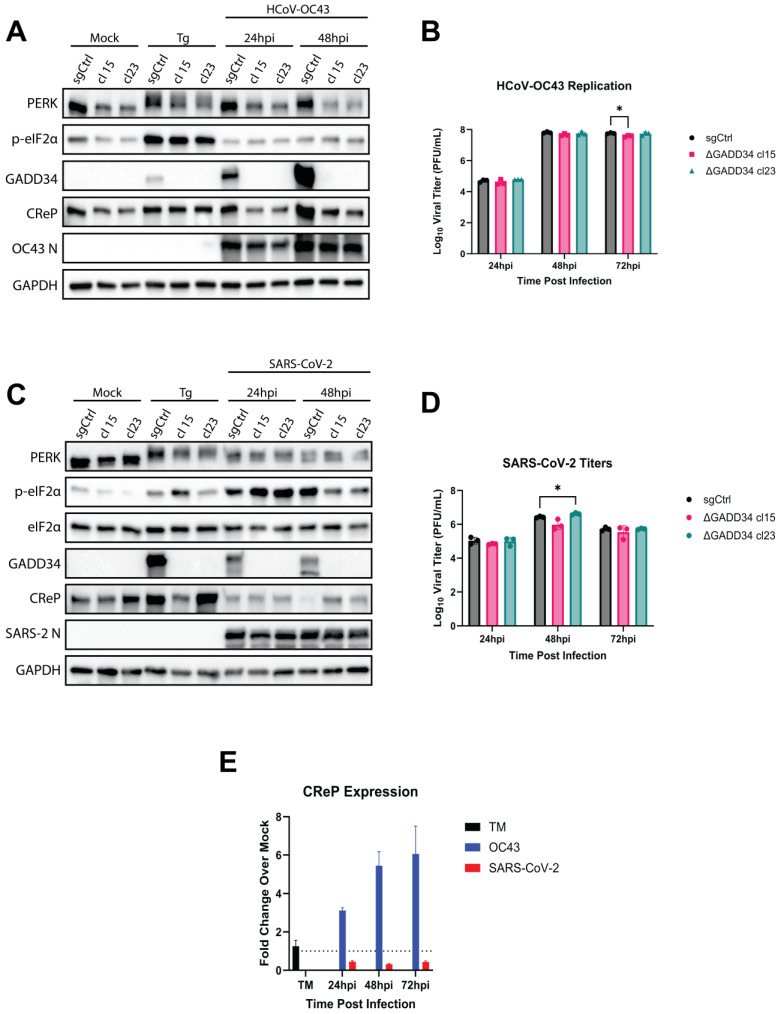
**GADD34 knockout has little impact on HCoV-OC43 and SARS-CoV-2 replication.** (**A**–**E**) Single-cell clones of A549^ACE2^ cells with either a nontargeting (sgCtrl) or GADD34 targeted sgRNAs were mock-infected or infected at MOI = 2 PFU/cell with HCoV-OC43 (**A**,**B**) or SARS-CoV-2 (**C**,**D**). At the indicated timepoints, whole-cell lysates (**A**,**C**) or RNA (**E**) were collected. (**A**,**B**) Western immunoblot for the indicated proteins. Thapsigargin (Tg, 1 h at 1 μM) was used as a positive control for ER stress. (**A**) HCoV-OC43 infections. (**C**) SARS-CoV-2 infections. (**B**,**D**) At the indicated timepoints, supernatants from infected cells were collected, and infectious virus titers quantified by plaque assay. (**B**) HCoV-OC43 infections. (**D**) SARS-CoV-2 infections. (**E**) Total RNA was used for RT-qPCR of CReP transcripts. Values are displayed as fold change over mock, calculated by 2^−Δ(ΔCt)^. Statistics by 2-way ANOVA. * = *p* < 0.05.

**Figure 7 viruses-17-00120-f007:**
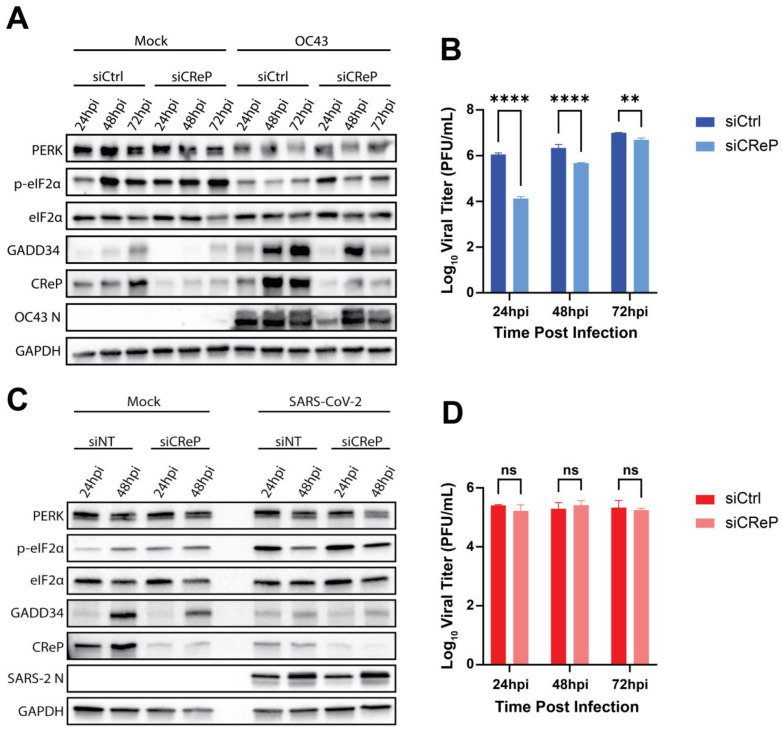
**CReP knockdown reduces HCoV-OC43, but not SARS-CoV-2, replication.** A549^ACE2^ cells were treated with siRNA targeting CReP (siCReP) or a scrambled control (siCtrl) for 72 h before mock-infection or infection with HCoV-OC43 (**A**,**B**) or SARS-CoV-2 (**C**,**D**) at an MOI = 2 PFU/cell. At the indicated timepoints, whole-cell lysates (**A**,**C**) or supernatants from infected cells (**B**,**D**) were collected. (**A**,**C**) Western immunoblots for the indicated proteins from HCoV-OC43-infected cells (**A**) or SARS-CoV-2-infected cells (**C**). (**B**,**D**) Viral titers were quantified by plaque assay for HCoV-OC43 (**B**) or SARS-CoV-2 (**D**) in the indicated conditions. Statistics by 2-way ANOVA. ** = *p* < 0.01; **** = *p* < 0.0001; ns = not significant.

**Figure 8 viruses-17-00120-f008:**
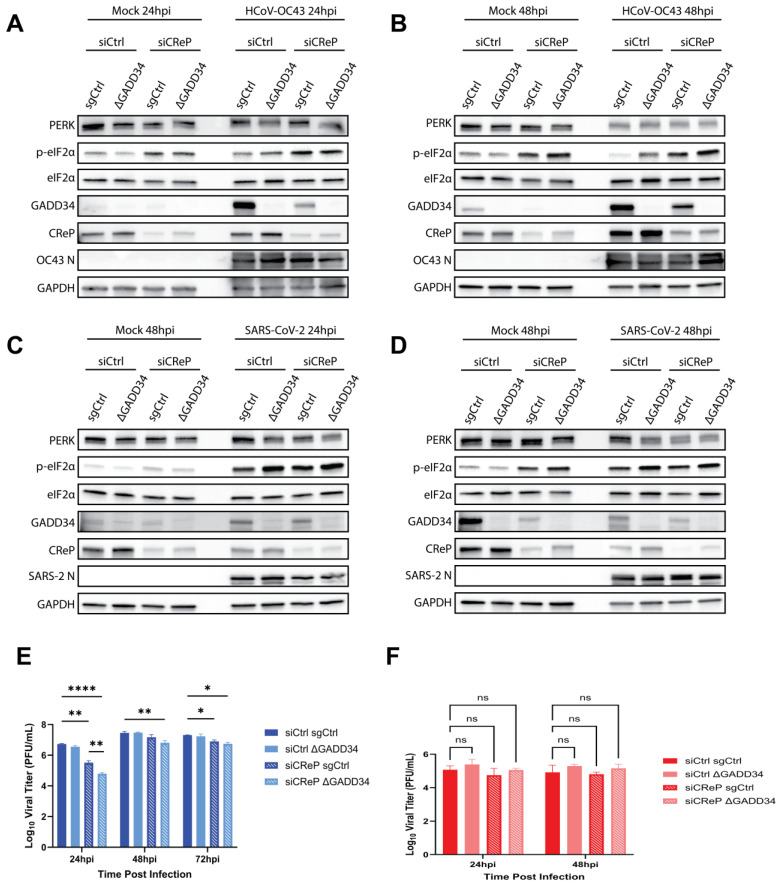
**CReP knockdown in GADD34-knockout cells has a combinatorial effect on HCoV-OC43 replication.** A549^ACE2^ sgCtrl cells or GADD34-KO cells (clone 23—ΔGADD34) were treated with control siRNA (siCtrl) or CReP-targeting siRNA (siCReP) for 72 h before being infected with HCoV-OC43 (**A**,**B**,**E**) or SARS-CoV-2 (**C**,**D**,**F**) at MOI = 2 PFU/cell. At the indicated timepoints, whole-cell lysates (**A**–**D**) or supernatants (**E**,**F**) were collected. (**A**–**D**) Western immunoblots for the indicated proteins were performed from HCoV-OC43-infected cells (**A**,**B**) or SARS-CoV-2-infected cells (**C**,**D**). (**E**,**F**) Infectious virus was quantified by plaque assay from HCoV-OC43-infected samples (**E**) and SARS-CoV-2-infected samples (**F**). Solid lines indicate siCtrl treatment, while dashed lines represent siCReP treatment. Statistics by 2-way ANOVA. * = *p* < 0.05; ** = *p* < 0.01; **** = *p* < 0.0001; ns = not significant.

## Data Availability

RNA-seq data have been deposited in the Gene Expression Omnibus database (GSE253542) (96). All other data are included in the manuscript and/or Appendix A.

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
