# Peer review of "Betacoronaviruses Differentially Activate the Integrated Stress Response to Optimize Viral Replication in Lung-Derived Cell Lines"

_viruses, 2025, doi:10.3390/v17010120_

Round 1
Reviewer 1 Report
Comments and Suggestions for Authors
This manuscript by Renner and coworkers presents an in-depth study investigating how betacoronaviruses regulate the host integrated stress response (ISR) to benefit viral infection. The host ISR system detects and responds to various forms of stress, including viral infections, and limits translation while attempting to restore homeostasis. However, many viral pathogens exploit this response to favor viral infection. The authors first analyzed previously published transcriptomic RNA-seq datasets and found that HCoV-OC43 and MERS-CoV significantly activated the IRS compared to SARS-CoV-2. They showed that all three viruses activate PERK and upregulate GADD34 during infection, but only SARS-CoV-2 induces p-eIF2a. Through chemical inhibition and genetic ablation methods, they show OC43 and MERS-CoV upregulate CReP to dephosphorylate eIF2a, which benefits viral replication. In contrast, SARS-CoV-2 prevents eIF2a dephosphorylation by limiting CReP and GADD34 expression.
Overall, this study is interesting and of high importance. It reveals the differential host ISR during three human respiratory betacoronaviruses and advances the understanding of viral pathogenesis. This study also provides insights into targeting the host ISR as a therapeutic strategy against some coronaviruses. The study was well-designed and carried out with proper controls. The manuscript is well-written, and the Discussion is comprehensive and inspiring. I have only a few very minor comments.
Minors:
Line 163: GADD34 should be non-italic.
Line 288: This section title may not be appropriate since the data suggest salubrinal or Sal003 inhibits the PP1 holoenzyme in a complex with CReP, not GADD34.
Line 575: There are two “merbecovirus”, change one to embecovirus
Reviewer 2 Report
Comments and Suggestions for Authors
The authors investigated the PERK pathway in the context of HCOV-OC43, MERS-CoV, and SARS-CoV-2 infections. They discovered distinct regulatory mechanisms, noting that only SARS-CoV-2 activated p-eIF2α, whereas the other two viruses did not. Furthermore, treatment with an eIF2α inhibitor revealed varying inhibitory effects across the three viruses. Subsequent gene knockout or knockdown experiments highlighted the potential roles of GADD34 and CREP in viral infections.
However, some concerns need to be addressed:
1. Line 153, add a citation.
2. Line 248–250, discuss GADD34 expression does not affect p-eIF2α levels during SARS-CoV-2 infection.
3. Line 262–266, the description in this section is unclear. Ensure the results are accurately correspond to the data in the figure.
4. Figure 6B and 6D, the two KO clones exhibit different levels of significance. The data should be reorganized for better clarity and consistency.
